# Enhancement of short/medium-range order and thermal conductivity in ultrahard $sp^3$ amorphous carbon by $C_{70}$ precursor

Yuchen Shang[1], Mingguang Yao [1] ✉, Zhaodong Liu[1,2], Rong Fu[3,4], Longbiao Yan[5], Long Yang[6], Zhongyin Zhang[7], Jiajun Dong[1], Chunguang Zhai[1], Xuyuan Hou[1], Liting Fei[5], GuanJie Zhang[5], Jianfeng Ji[5], Jie Zhu[7], He Lin [5], Bertil Sundqvist[8] & Bingbing Liu [1,2] ✉

As an advanced amorphous material, $sp^3$ amorphous carbon exhibits exceptional mechanical, thermal and optical properties, but it cannot be synthesized by using traditional processes such as fast cooling liquid carbon and an efficient strategy to tune its structure and properties is thus lacking. Here we show that the structures and physical properties of $sp^3$ amorphous carbon can be modified by changing the concentration of carbon pentagons and hexagons in the fullerene precursor from the topological transition point of view. A highly transparent, nearly pure $sp^3$−hybridized bulk amorphous carbon, which inherits more hexagonal-diamond structural feature, was synthesized from $C_{70}$ at high pressure and high temperature. This amorphous carbon shows more hexagonal-diamond-like clusters, stronger short/medium-range structural order, and significantly enhanced thermal conductivity ($36.3 \pm 2.2$ W m$^{-1}$ K$^{-1}$) and higher hardness ($109.8 \pm 5.6$ GPa) compared to that synthesized from $C_{60}$. Our work thus provides a valid strategy to modify the microstructure of amorphous solids for desirable properties.

It is well known that the microstructure of materials determines their properties. For crystalline solids, the long-range ordered arrangement of atoms allows a precise characterization of the atomic-level structures and also an establishment of the corresponding structure-properties relationships[1–4]. In contrast, because of the absence of long-range order, characterization of the atomic-level structure of non-crystalline materials is difficult and to build the structure-property relationships remains challenging[5,6]. Despite the absence of long-range order, non-crystalline solids often contain short- and medium-range order (SRO, MRO) at various length scales (1–20 Å)[7,8]. Great efforts have been made to study how these develop and distribute in amorphous structures and how they influence the macroscopic properties of amorphous materials (e.g., bulk metallic glasses (BMG) or amorphous silicon (a-Si))[9–14]. Several approaches have been used to tune the internal local structural order in amorphous solids by changing the processing history, such as annealing, high pressure treatment, or quenching by different cooling rates[15–21].

Carbon can form a variety of allotropes with dramatically different properties, because of the flexibility in forming $sp^1$−, $sp^2$−, and $sp^3$−hybridized bonds[22]. Among which, $sp^3$−hybridized carbon

[1]State Key Laboratory of Superhard Materials, College of Physics, Jilin University, Changchun 130012, China. [2]Synergetic Extreme Condition User Facility, Jilin University, Changchun 130012, China. [3]School of Materials Science and Engineering, State Key Laboratory of Advanced Special Steel, Shanghai University, Shanghai 200444, China. [4]School of Mathematical and Physical Sciences, University of Technology Sydney, New South Wales 2007, Australia. [5]Shanghai Synchrotron Radiation Facility, Shanghai Institute of Applied Physics, Chinese Academy of Sciences, Shanghai 201204, China. [6]School of Materials Science and Engineering, Tongji University, Shanghai 201804, China. [7]School of Energy and Power Engineering, Key Laboratory of Ocean Energy Utilization and Energy Conservation of Ministry of Education, Dalian University of Technology, Dalian 116024, China. [8]Department of Physics, Umeå University, SE-90187 Umeå, Sweden. ✉e-mail: yaomg@jlu.edu.cn; liubb@jlu.edu.cn

materials have been attracting particular interest because of their extraordinary properties. The well-known crystalline forms of $sp^3$– carbon such as cubic diamond (CD)[23], hexagonal diamond (HD)[24], and monoclinic carbon (such as M-carbon, V-carbon, etc.)[25,26] possess excellent mechanical properties and other physical properties, depending on the structures. For example, diamond is the hardest material known in nature while the hexagonal form is much harder than that of the cubic one[27]. Considering that amorphous material could inherit SRO and MRO from the corresponding crystal, $sp^3$ amorphous carbon is expected to exhibit various structure, which could preserve some of its properties similar to the crystal while still exhibit new properties. Unfortunately, because of the very high melting point of diamond, above 4500 K, $sp^3$–hybridized bulk amorphous diamond cannot be synthesized by quenching liquid carbon[28]. So, it is not feasible to tune the local structural order by using the traditional processes that are applied to metallic glass, for example. Thus, it is necessary to explore new ways to control the microstructures of $sp^3$ amorphous carbon in order to improve its properties and expand the range of possible applications.

Very recently, we successfully synthesized millimeter-sized samples of bulk amorphous carbon with nearly pure $sp^3$–hybridized bonds by compressing fullerene $C_{60}$ at high pressure and high temperature (HPHT)[29]. The material exhibits exceptional mechanical, thermal, and optical properties beyond those of all other known amorphous solids. From a topological transition point of view, the presence of a large number of pentagons in the starting precursor $C_{60}$ is suggested to be crucial for the formation of amorphous structures[30]. In this case, the local structural order in $sp^3$ amorphous carbon is expected to be tuned by changing the concentration of carbon hexagons and pentagons in the precursor. We note that $C_{70}$, the next higher fullerene, contains five more carbon hexagons than $C_{60}$. Also, it has only ten reactive double bonds localized in the pentagonal "polar cap" parts of the molecule while

the graphene-like ribbon, consisting of hexagons, localized around the "equator" is rather inert[31–33]. Moreover, due to the lower symmetry and reactivity, $C_{70}$ exhibits a different compression behavior under pressure compared to that of $C_{60}$, such as higher stability and anisotropic deformation and polymerization[34–37], making it more difficult to transform into ordered crystalline structures. These differences might have a significant impact on the formation of amorphous phases under high pressure.

Motivated by this, we have investigated the transformation of $C_{70}$ fullerene into $sp^3$ amorphous carbon under pressures of 18–30 GPa and in a temperature range of 900–1200 °C, using a large-volume press. A highly transparent, nearly completely $sp^3$–hybridized bulk amorphous carbon with increased hexagonal-diamond-like clusters and strong local structural order was synthesized at 30 GPa and 1100 °C. The strong local structural order in the $sp^3$ amorphous carbon should be related to the increased concentration of carbon hexagons in the fullerene precursor, which also contributes to a significantly enhanced thermal conductivity and higher hardness.

## Results

### Sample synthesis and structure

Sublimed $C_{70}$ powders were used as starting precursors for HPHT synthesis in this study. High-quality, millimeter-sized samples were obtained after HPHT treatment at pressures from 18 to 30 GPa and temperatures from 900 to 1200 °C. Figure 1 shows optical images and X-ray diffraction (XRD) patterns of the recovered samples. The macroscopic morphology of samples changed significantly with increasing pressure and temperature, from black opaque to yellow translucent and finally nearly colorless transparent (Fig. 1b). XRD results show that the starting $C_{70}$ material is crystallized in a rhombohedral structure, as previously reported by Marques et al.[36,37], while the samples recovered from HPHT treatment exhibit completely different patterns with two very broad diffraction peaks around 42° and 84°, indicating an

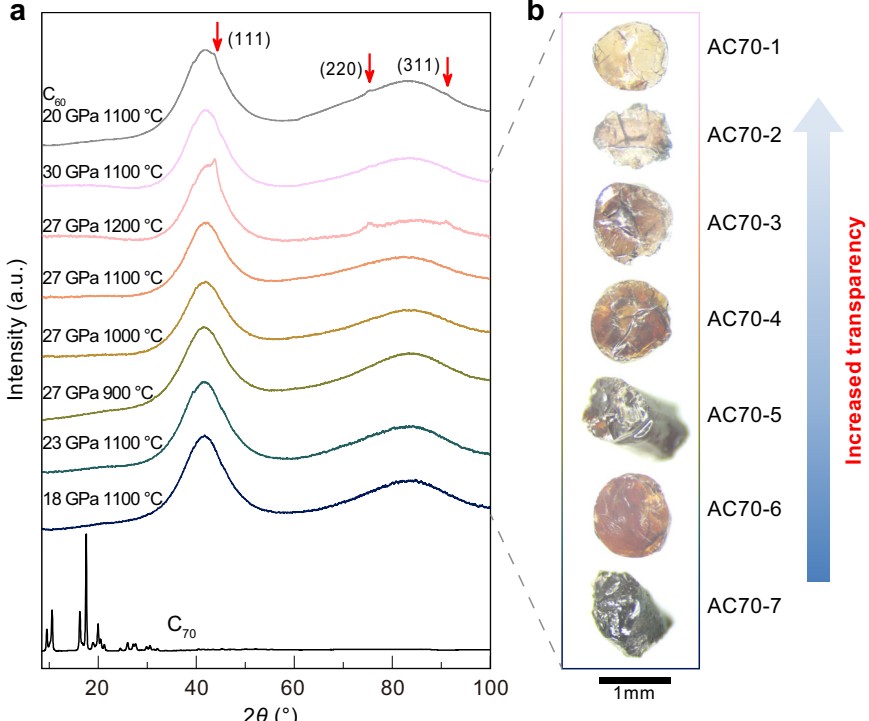

**Fig. 1 | XRD patterns and optical images of $C_{70}$ samples recovered from different P-T conditions. a** XRD patterns of $sp^3$ amorphous carbon samples recovered from different P-T conditions. For comparison, XRD results for a $C_{60}$ sample treated under 20 GPa and 1100 °C (ref. 29) are shown (top curve). And the XRD patterns of pristine $C_{70}$ are also shown at the bottom. a. u. arbitrary units. **b** The optical images of $sp^3$ amorphous carbon samples recovered from different P-T conditions. The transparency of the samples is improved with the increasing P-T conditions.

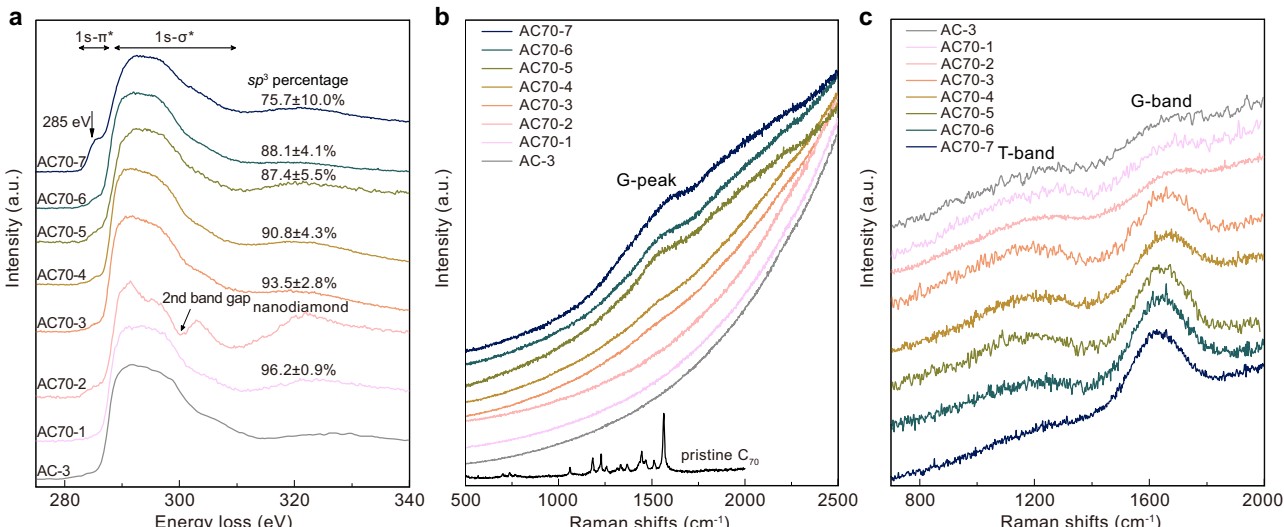

**Fig. 2 | Bonding state and $sp^3$ content of recovered samples.** Data from EELS spectra (**a**) Visible (514 nm) (**b**) and UV (325 nm) (**c**) Raman spectroscopy of samples recovered from different HPHT conditions. **a** The lower energy peak at approximately 285 eV represents the π-bonding feature, corresponding 1s to π* transition (labeled 1s-π*), and the broad band at higher energy features the σ-bonding, corresponding 1s to σ* transition (labeled 1s-σ*). The error bars of $sp^3$ percentage indicate three different measured points, standard deviations. The spectrum of AC70-2 exhibits typical spectroscopic features of nanodiamond, in which a dip at -302 eV is attributed to the second absolute bandgap of diamond (2nd bandgap). The Raman spectrum of pristine $C_{70}$ is also shown in **b** (514 nm excitation). AC-3 is the nearly pure $sp^3$ amorphous carbon synthesized from $C_{60}$ (ref. 29). a. u. arbitrary units.

amorphous structure (Fig. 1a). The amorphous carbon samples synthesized from $C_{70}$ are labeled as AC70-1 to AC70-7, respectively (Fig. 1b). Note that increasing the temperature to 1200 °C at 27 GPa resulted in the presence of very weak diffraction peaks from cubic diamond, suggesting the formation of small amounts of crystalline diamond in the AC70-2 sample at higher temperature (Fig. 1a). In contrast, the presence of cubic diamond was found at lower *P-T* conditions (20 GPa 1100 °C) in the case of using a $C_{60}$ precursor.

## Bonding features and $sp^3$ content

Carbon K-edge electron energy loss spectroscopy (EELS) was used to examine the bonding state and measure the $sp^3$ content of our amorphous carbon (Fig. 2a). We can see that the 1s-π* peak at 285 eV, corresponding to $sp^2$ carbon, gradually decreases with increasing *P-T* conditions, and is finally absent for the AC70-1 sample recovered from 30 GPa and 1100 °C. Using $sp^2$ glassy carbon as the standard material, the percentage of $sp^3$−hybridized carbon is quantified as high as 96.2 ± 0.9% from a calculation of the ratio of integrated areas under the π* and σ* peaks[29,38]. The results indicate that $C_{70}$ transforms into nearly pure $sp^3$ amorphous carbon at 30 GPa and 1100 °C.

This nearly pure $sp^3$ amorphous carbon was further confirmed by our Raman measurements. Figure 2b, c show the visible and ultraviolet (UV) Raman spectra of samples recovered from different HPHT conditions. For the samples recovered from 18-27 GPa and temperatures of 900–1100 °C, the visible Raman spectrum exhibits a broad G peak at around 1560 cm$^{-1}$ representing characteristics of $sp^2$− carbon (Fig. 2b), and the peak intensity gradually weakens as the *P-T* conditions increases. The G peak was not detected in the samples synthesized at 30 GPa and 1100 °C (AC70-1) or 27 GPa and 1200 °C (AC70-2), indicating that $sp^2$− carbon nearly disappeared in these products. Compared to visible light excitation, UV Raman is more sensitive to the vibrations of $sp^3$− carbon in carbon materials[39,40]. As shown in Fig. 2c, besides the G-band from $sp^2$− carbon, a T-band at -1100 cm$^{-1}$ attributed to $sp^3$− carbon is observed in the UV Raman spectra for all the samples. The ratio between the intensities of the two peaks ($I_T/I_G$) increases with increasing *P-T* conditions, and the $I_T/I_G$ reaches a maximum for the AC70-1 sample recovered from 30 GPa and 1100 °C, indicating that its $sp^3$ percentage is the highest. To our knowledge, the maximum value

found here is higher than in any other known amorphous carbon material.

## Microstructure

Figure 3a–c show typical high-resolution transmission electron microscopy (HRTEM) images and selected area electron diffraction (SAED) patterns of three representative samples recovered from different HPHT conditions. The HRTEM image of AC70-1 exhibits a maze-like pattern with no discernible long-range ordered structure (Fig. 3a). The collected SAED pattern also displays two diffuse halos, corresponding to the two broad peaks in XRD (Figs. 1a and 3a). In contrast, HRTEM observations reveal that a small amount of a disordered graphite structure coexists with amorphous carbon in the AC70-7 sample synthesized at a lower pressure of 18 GPa, 1100 °C (Fig. 3c), while for the AC70-2 sample obtained at 27 GPa and 1200 °C, nanocrystalline diamond is clearly observed (Fig. 3b). These observations are consistent with the above XRD and EELS results.

The local structural order of the AC70-1 sample was further characterized by inverse fast Fourier transform (FFT) of HRTEM images. Figure 3d shows the local structure heterogeneity down to the range of 5-20 Å. On this scale, many diamond-like MRO clusters can be observed, randomly distributed in the amorphous matrix. Similar structural characteristics were also observed in other amorphous solids with typical MRO structures, such as amorphous $SiO_2$ (a-$SiO_2$), BMG, and $sp^3$−hybridized amorphous carbon from $C_{60}$[29,41–43]. These clusters can be identified and distinguished as having two types of lattice-like fringes with different intersecting angles, corresponding to the (111) plane of cubic diamond and the (100) plane of hexagonal diamond (Fig. 3d, e, Supplementary Fig. 1a, b). To further quantify the atomic structural order, statistical analysis was performed in three randomly selected areas (Supplementary Fig. 1d–f). The results show that the total areal fraction of the local ordered regions is 38.1 ± 3.8% (Fig. 3f, Supplementary Fig. 1c). For comparison, we made a similar analysis on sample AC-3, synthesized from $C_{60}$. The total areal fraction of 31.6 ± 2.3% for AC-3 is lower than that of the AC70−1 sample from $C_{70}$ (Fig. 3e, f, Supplementary Fig. 1g–i). In addition, we also counted the percentage of CD and HD-like clusters in the amorphous carbon and found that the fraction of HD-like clusters in AC70-1 is about twice that

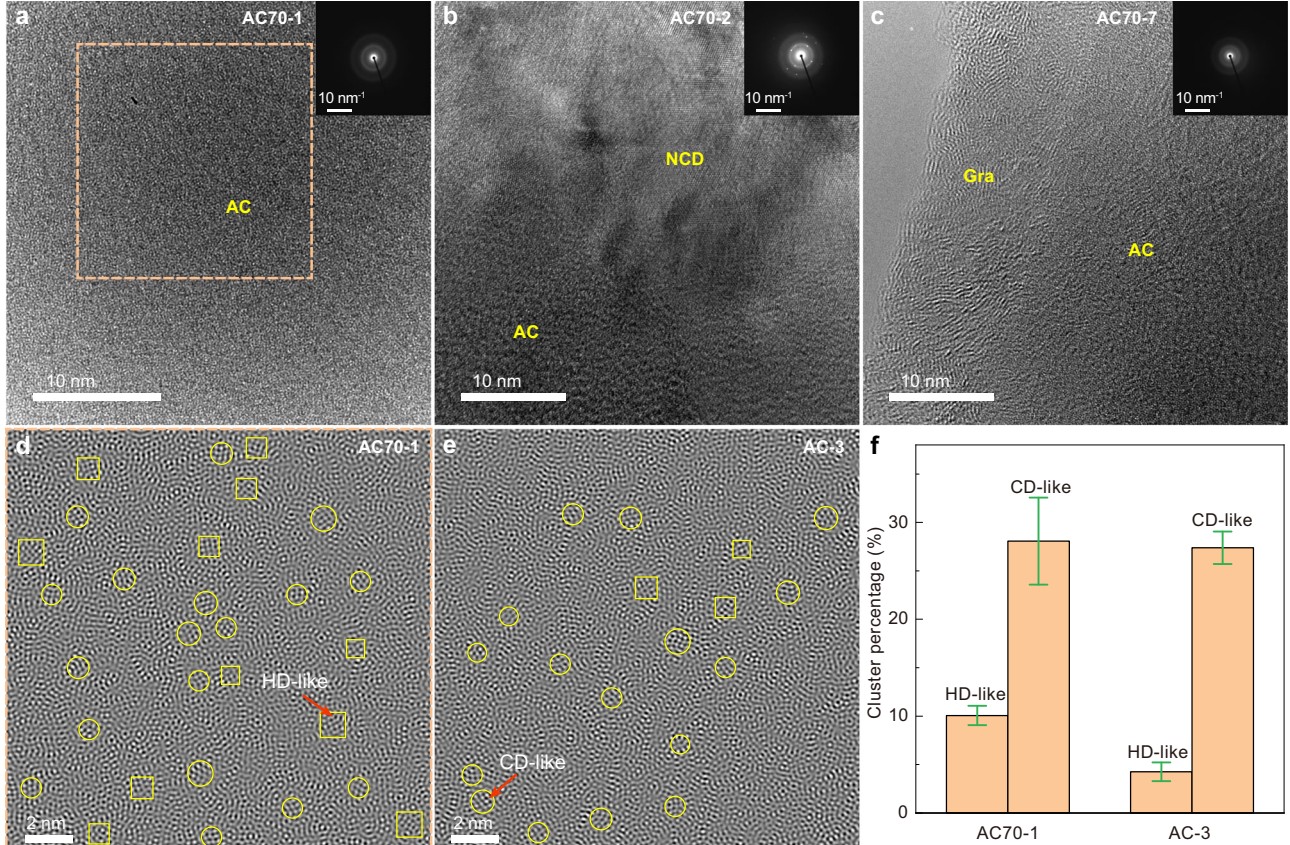

**Fig. 3 | TEM characterization of samples recovered from different *P-T* conditions. a**–**c** HRTEM images of samples recovered from 30 GPa and 1100 °C (AC70-1) (**a**) 27 GPa and 1200 °C (AC70-2) (**b**) and 18 GPa and 1100 °C (AC70-7) (**c**). Inserts show the corresponding SAED patterns collected from the same regions. **d** The inverse FFT image of the area marked with an orange box in (**a**). **e** The inverse FFT image of *sp*³ amorphous carbon (AC-3) synthesized from C₆₀ (ref. 29). The regions marked with yellow circles and squares in **d** and **e** represent the CD-like and HD-like clusters, respectively. **f** Content statistics of diamond-like MRO clusters in $sp^3$ amorphous carbon samples synthesized from $C_{60}$ and $C_{70}$. Error bars indicate statistical analysis for three different areas, standard deviations. Gra disordered graphite, AC amorphous carbon, NCD nanocrystalline diamond.

in AC-3 (Fig. 3f, Supplementary Fig. 1c). These results confirm that the local structural order of $sp^3$ amorphous carbon can be tuned by using different fullerene precursors.

### Local MRO structure and thermal transport properties

The short- and medium-range structure of $sp^3$ amorphous carbon was further characterized by analyzing the structure factor and pair distribution function (PDF). Figure 4a shows the structure factors $S(Q)$ of amorphous carbon samples obtained from $C_{70}$ and $C_{60}$ at HPHT. Two intense diffraction peaks occur in the low-$q$ range. Previous studies show that the low-$q$ peak structure in $S(Q)$ is due to the correlation between clusters (MRO)[44]. The former $Q_1$, usually called the first sharp diffraction peak (FSDP), has often been observed in many amorphous materials like a-Si, silica, and BMG[41,44,45]. A generally accepted view for the structural origin of FSDP is that it is primarily associated with the presence of MRO in amorphous materials and the intensity of FSDP has a positive correlation with the local structural order[8,21,41,46,47]. As shown in Fig. 4a, the intensity of the FSDP of $sp^3$ amorphous carbon synthesized from $C_{70}$ is obviously higher than that of the AC-3 sample synthesized from $C_{60}$, indicating a higher local structural order in AC70-1 than that in AC-3. Figure 4b provides a magnified view of the FSDP in $S(Q)$, revealing that the peak position of FSDP for AC70-1 is clearly higher than that of AC-3. This indicates a slightly increased density for AC70-1. The difference in local structural order was further confirmed by analyzing the real-space structure information. The reduced PDF, $G(r)$, obtained by Fourier transform of the structure factors, is presented in Fig. 4c, d. The peak

intensity of the first shell $r_1$ and second shell $r_2$ are both stronger for AC70-1. This confirms that the local structural order of $sp^3$ amorphous carbon synthesized from $C_{70}$ is higher than that from $C_{60}$. Besides, it is also found that the first $r_1$ and second $r_2$ peak for AC70-1 both shift slightly to the left and are closer to the standard tetrahedral atomic configuration of crystalline diamond ($r_1$, 1.54 Å, $r_2$, 2.515 Å) (Fig. 4d). These results further prove the higher density and local structural order for AC70-1, which is consistent with the results of $S(Q)$.

The short-range atomic configuration was investigated by analyzing the atomic PDF $g(r)$, which is transformed from $G(r)$ ($g(r) = 1 + G(r)/(4\pi r\rho_0)$, where $\rho_0$ is the average number density). Figure 4e shows the atomic PDF $g(r)$ of AC70-1 sample. The first and second peak positions of $g(r)$, corresponding to the average nearest- and next-nearest-neighbor distances of $sp^3$ amorphous carbon, is 1.55 Å and 2.52 Å, respectively. The average C-C-C bond angle was further calculated to be 108.8° from the $r_1$ and $r_2$ [$\theta = 2\sin^{-1}(r_2/2r_1)$] and the average coordination number was estimated to be ~4.05 from the first peak area. All the atomic structure information indicates the formation of a short-range tetrahedral structure, close to that of crystalline diamond. Note that our $sp^3$ amorphous carbon exhibits obvious differences in $g(r)$ compared with paracrystalline diamond (P-D) (Fig. 4e), a composite consisting of an amorphous carbon matrix filled with different percentages of nanometer-scale, severely distorted crystals[30,47]. The P-D exhibits two additional peaks in the long-range region, but such local paracrystalline structural order is absent in our $sp^3$ amorphous carbon.

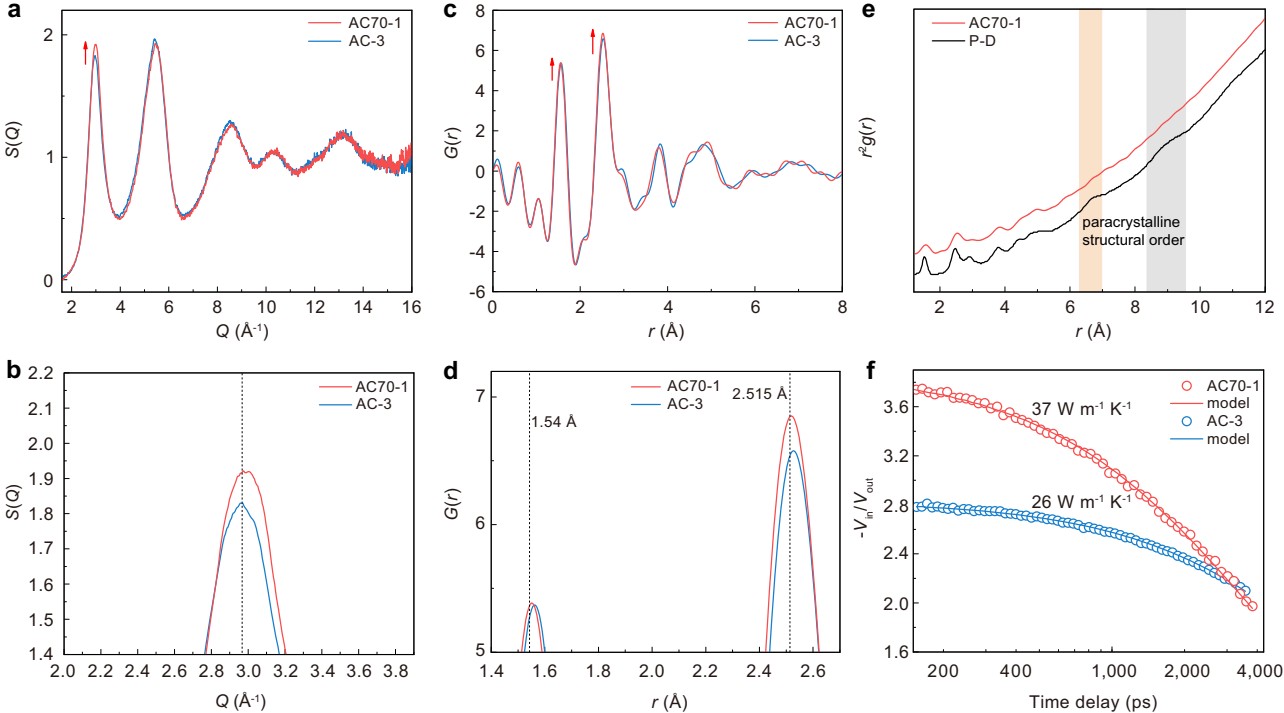

**Fig. 4 | Atomic-scale structure and thermal conductivity of AC70-1 and AC-3.**
**a** Structure factor $S(Q)$ of AC70-1 and AC-3 samples. **b** The magnified FSDP in $S(Q)$. The dashed line represents the peak position of FSDP for AC-3, which is slightly lower than that of AC70-1. **c** The reduced PDF profiles, $G(r)$, of AC70-1 and AC-3 samples. The red arrows in **a** and **c** indicate peak intensity increase for AC70-1. **d** The magnified first $r_1$ and second $r_2$ peak in $G(r)$. The two dashed lines represent the standard first and second nearest-neighbor atomic distances of crystalline

diamond, respectively. **e** The atomic PDF, $g(r)$, of the AC70-1 sample and para-crystalline diamond (ref. [47]). Two additional peaks in the long-range region marked by the orange and gray sections are the typical local paracrystalline structural order of P-D. The $g(r)$ curve for AC70-1 was shifted vertically for comparison. **f** TDTR data of AC70-1 and AC-3 samples. The solid lines represent the best fits to the thermal model.

The local structural order in amorphous carbon could also significantly affect the thermal conductivity. Figure 4f shows the representative time-domain thermoreflectance (TDTR) ratio signals of AC70-1 and AC-3 samples. Although the two samples contain similar $sp^3$ content, 96.2 ± 0.9% for AC70-1 and 95.1% ± 1.7% for AC-3, it is found that AC70-1 sample exhibits a significantly enhanced thermal conductivity of 36.3 ± 2.2 W m$^{-1}$ K$^{-1}$ compared to that in AC-3 synthesized from $C_{60}$ (26.0 ± 1.3 W m$^{-1}$ K$^{-1}$) (Supplementary Figs. 4, 5)[29]. Previous studies revealed that the thermal conductivity of amorphous carbon is mostly governed by the content and structural order of the $sp^3$ phase[48,49]. This indicates that the higher thermal conductivity in amorphous carbon from $C_{70}$ should originate from the different internal local structural order compared with that from $C_{60}$.

### Mechanical and optical properties

The $sp^3$ amorphous carbon synthesized from $C_{70}$ exhibits superior mechanical properties and tunable optical bandgaps. The hardness-load curve of AC70-1 (Fig. 5a) shows that the hardness reaches an asymptotic value at a load of 4.9 N and it exhibited a Vickers hardness of 109.8 ± 5.6 GPa at the maximum load of 9.8 N. This is higher than that of AC-3 synthesized from $C_{60}$. The optical bandgap of the amorphous carbon was measured by UV-visible light absorption. As shown in Fig. 5b, the cut-off edges of the absorption spectra recorded from the samples are blue-shifted with increasing $P$-$T$ synthesis conditions. The optical bandgap was further calculated by plotting $(\alpha h\nu)^{1/2}$ as a function of photon energy (Fig. 5c), where $\alpha$, $h$, and $\nu$ are the optical absorption coefficient, Planck's constant and the photon frequency, respectively[50]. The AC70-1 sample exhibits the largest bandgap of 2.80 eV, which is also slightly wider than that of AC-3. We also plot the variation of optical bandgap against $sp^3$ content for our $sp^3$ amorphous carbon and other amorphous carbon materials[51] (Fig. 6a). It can be

seen that the bandgaps of our amorphous carbon materials show a strong correlation with their $sp^3$ content and that the dependence is also consistent with data for other amorphous carbons[51].

## Discussion

The millimeter size of the amorphous carbon samples produced makes it possible to carry out a comprehensive study on their structures, mechanical and thermal transport properties. The nearly pure $sp^3$ amorphous carbon inherits more hexagonal-diamond structural feature and shows higher local structural order than that synthesized from $C_{60}$, which exhibits a much higher thermal conductivity and a higher Vickers hardness at a load of 9.8 N (Fig. 6c) than any other amorphous solid known up to now. Note that thermal transport properties in diamond-like amorphous carbon film are mainly governed by the $sp^3$ content and the structural order of the $sp^3$ phase[49]. The $sp^3$ amorphous carbon synthesized from $C_{70}$ exhibits a high thermal conductivity of 36.3 ± 2.2 W m$^{-1}$ K$^{-1}$, to the best of our knowledge this is the higher reported value for an amorphous material. This is also much higher than for the $sp^3$ amorphous carbon (26.0 ± 1.3 W m$^{-1}$ K$^{-1}$) with similar $sp^3$ content synthesized from $C_{60}$ (Fig. 6b, c). In amorphous solids, mechanical vibrations carrying heat are divided into propagating propagons, diffusing diffusons and localized locons[52]. Among them, the low-frequency propagons, resembling phonons in crystalline solids, have relatively large mean free paths and high heat transfer efficiency[52]. Previous theoretical simulation results have shown that an amorphous material with higher MRO exhibits larger thermal conductivity as the propagon thermal conductivity in a MRO structure is larger than that in an ideal amorphous structure[13]. This is the reason why the current $sp^3$ amorphous carbon with higher local structural order exhibits such a high thermal conductivity.

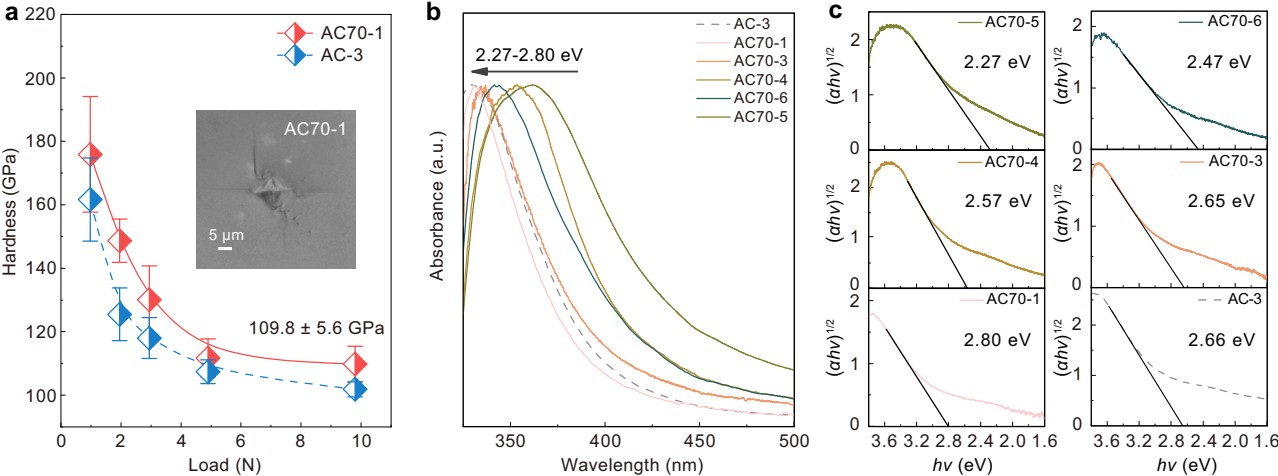

**Fig. 5 | Hardness and optical bandgap of $sp^3$ amorphous carbon synthesized from $C_{70}$. a** Vickers hardness of AC70-1 and AC-3 samples as a function of applied load force. Inset is the optical micrograph of the indentation after 9.8 N load; Error bars indicate five different measurement points, standard deviations. **b** UV-visible absorption spectra of samples recovered from different HPHT conditions. a. u. arbitrary units. **c** Plots of $(\alpha h\nu)^{1/2}$ *vs* the photon energy of $sp^3$ amorphous carbon samples. The black slash lines indicate the cut-off edges of the spectra.

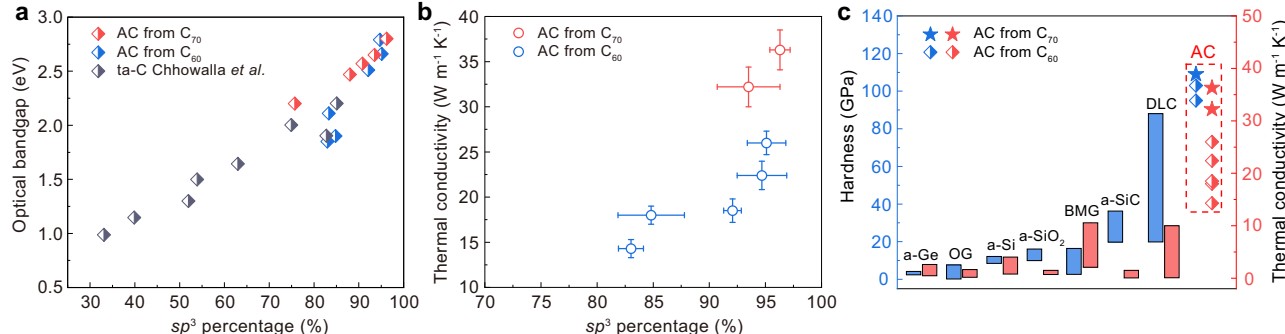

**Fig. 6 | The physical properties of $sp^3$ amorphous carbon synthesized from $C_{70}$ compared with other amorphous materials. a** Optical bandgaps of $sp^3$ amorphous carbon and tetrahedral amorphous carbon (ta-C) (ref. [51]) as a function of $sp^3$ content. **b** The thermal conductivities of amorphous carbon materials as a function of $sp^3$ content. Error bars indicate three different measurement points, standard deviations. **c** The hardness and thermal conductivity of $sp^3$ amorphous carbon in comparison with other known amorphous materials. Physical properties of amorphous carbon synthesized from $C_{60}$ were obtained from our previous work (ref. [29]). a-Ge amorphous Ge, OG oxide glasses, a-SiC amorphous SiC, DLC Diamond-like carbon film.

The unique configuration of the $C_{70}$ precursor should be responsible for the amorphous structure formed, and for its superior properties. As we know, fullerene cages collapse under pressure, accompanied with $sp^2$–$sp^3$ bonding changes[53,54]. In this case, from the topological transition point of view, the presence of both hexagons and pentagons makes formation of an ordered structure unlikely[30], but overheating could induce re-crystallization and diamond formation, as observed in our comparison experiments. $C_{70}$ contains five more carbon hexagons than $C_{60}$ and these connected hexagons form a graphene-like ribbon at the equatorial belt of the ellipsoidal molecule. This ribbon is rather inert compared to the high-curvature caps consisting of carbon pentagons and hexagons. Previous studies have shown that crystalline $C_{70}$ transforms into a one-dimensional zig-zag chain structure or a two-dimensional layered structure through anisotropic polymerization via molecular cap linking at suitable HPHT conditions[35–37]. In these polymerized phases, the graphene-like ribbons at the equatorial belt of $C_{70}$ are retained. According to the proposed mechanisms for the graphite-to-diamond transition under HPHT, i.e., either a concerted mechanism or a nucleation mechanism, cubic or hexagonal diamond could be formed by collective sliding, bending and buckling of graphitic layers with a purely hexagonal structure[55–57]. The graphene-like ribbons in

$C_{70}$ should favor the transformation into diamond-like MRO under HPHT, leading to a higher local structural order with a higher fraction of hexagonal-diamond-like clusters in the obtained amorphous carbon compared with using $C_{60}$.

In summary, a millimeter-sized, highly transparent, and nearly pure $sp^3$ amorphous carbon sample with excellent mechanical and thermal transport properties was synthesized from $C_{70}$ fullerene at a pressure of 30 GPa and high temperature. Compared to similar materials produced from $C_{60}$, this amorphous carbon inherits more hexagonal-diamond structural feature, showing a higher fraction of hexagonal-diamond-like clusters and stronger short/medium-range structural order, which should be related to the increased concentration of carbon hexagons in the fullerene precursor. Such microstructures also contribute to its significantly enhanced thermal conductivity and higher hardness. The structures and physical properties of $sp^3$ amorphous carbon thus can be modified by changing the concentration of carbon pentagons and hexagons in the precursor, providing a valid strategy to modify the local structural order of amorphous solids. New $sp^3$ amorphous carbons with different structural orders and superior physical properties are expected to be produced from larger fullerenes under HPHT. This strategy may also be extended to other covalently bonded amorphous solids, such as

 

amorphous boron nitride and silicon, to create desired amorphous materials.

## Methods

### Materials preparation

The starting sublimed $C_{70}$ powders were purchased from Tokyo Chemical Industry Co., Ltd (TCI). High pressure quench experiments at 18–30 GPa were performed by using a 7/3 (OEL/TEL = octahedral edge length of pressure medium/truncated edge length of anvil) cell assembly in a 10-MN Walker-type large-volume press at the State Key Laboratory of Superhard Materials, Jilin University[58]. $C_{70}$ powders were enclosed in a rhenium capsule (also acting as heater) and put into $ZrO_2$ sleeves in a $Cr_2O_3$-doped MgO octahedron. Pressure calibrations of the cell assembly at room temperature and high temperature were reported in our previous work[58]. Temperature was measured with a $W_{75}Re_{25}$-$W_{97}Re_3$ thermocouple adjacent to the sample capsules. The samples were first compressed to preset pressures in around 10 h at room temperature and then heated to the target temperatures with a heating rate of 100 °C min.$^{-1}$ and held for 1–2 h. After that, the target temperature was quenched (~ 500 °C s.$^{-1}$) by shutting off the heating power, followed by slow pressure release over the course of approximately 15 h.

### Structure characterization

The structures of the recovered samples were characterized by using an X-ray diffractometer (MicroMax-007HF, Rigaku, Japan) with Cu Kα radiation (wavelength, 1.5418 Å). Microstructures of the recovered samples were further characterized by a field-emission transmission electron microscope (TEM, JEM-2200FS JEOL) with an acceleration voltage of 200 kV. EELS spectra around the carbon K-edge were also collected in the TEM mode to determine the bonding states in the samples. Using glassy carbon as the standard purely $sp^2$ amorphous material, the $sp^3$ content of the recovered amorphous carbon samples were quantified by calculating the ratio of integrated areas under the 1s-π* and 1s-σ* peaks. The visible and UV Raman spectra of samples were collected at room temperature on a Renishaw inVia spectrometer with 514-nm excitation source and a Horiba-Jobin Yvon LabRAM ARAMIS system with 325-nm excitation source, respectively.

To obtain high-quality data for the structure factor $S(Q)$ and pair distribution function, rapid-acquisition pair distribution function (RAPDF) scattering measurements were performed at the BL13HB beamline of the Shanghai Synchrotron Radiation Facility (SSRF). The center energy of the X-ray is 39.99 keV radiated from a high field superbending (B = 2.3 T) magnet. The synthesized samples were enclosed in a boron-rich thin-walled capillary tube (1.5-mm outer diameter, 10-μm wall thickness) for XRD measurements and the data was collected for 8 s on a Perkin detector. The high-quality diffraction data with $Q_{max}$ up to 16 Å$^{-1}$ was collected by reducing the distance between the sample and detector. A background file was obtained by collecting empty capillary signals with the same collection condition. DIOPTAS[59] is used for calibrating and integrating the 2D signal. The environmental scattering, incoherent and multiple scattering, polarization, and absorption were correcting by using PDFGETX2[60] to obtain the high-quality PDF file. The average coordination number was obtained by integrating $4\pi r^2 \rho_0 g(r)$, where $\rho_0$ denotes the average atom number density. The density of the AC70-1 sample was determined as ~3.4 ± 0.1 g/cm$^3$ by a simple floating method[29]. Note that the peak intensity of $S(Q)$ and $G(r)$ is sensitive to the experimental conditions and data analysis, such as the experimental coverage of $Q$ or $Q$ cut-off and the background subtraction. In comparison, the peak position is usually more reliable. We performed two separate measurements on each sample at the BL13HB beamline of the SSRF (Fig. 4 and Supplementary Fig. 2), which give quite similar results in both peak intensities and positions of $S(Q)$ and $G(r)$ for the measured samples. For example, the two measurements show that the peak intensities of FSDP in $S(Q)$

and the first $r_1$ and second $r_2$ peak in $G(r)$ for AC70-1 are higher than those of AC-3.

### Physical property measurements

The indentation experiments were carried on the end-polished samples by using a microhardness tester (HV-1000ZDT) equipped with a four-sided pyramidal diamond indenter. The maximum applied load was 9.8 N, and the dwell time was 10 s. Five different measurement points were obtained at each load. The Vickers hardness value was determined according to the formula: $H_V = 1,854.4F/L_1^2$, where $F$ (N) is the applied load and $L_1$ (μm) is the arithmetic mean of the two diagonals of the Vickers indentation.

The optical bandgap of samples was measured by UV-visible light absorption. UV-visible absorption spectra were recorded on a UV-visible spectrometer (iHR320).

The thermal transport properties of the recovered samples were measured by noncontact TDTR methods based on a pump-probe technique[61] (Supplementary note 1). In TDTR measurements, a mode-locked Ti: sapphire laser serves as the light source, which produces a train of pulses (~280 fs) at a repetition rate of 80 MHz. A polarizing beamsplitter divides the laser into a pump beam and a probe beam. The laser power is 20 mW for the pump and 10 mW for the probe. A mechanical delay stage varies the optical path of the pump beam, producing a time delay of up to ~4 ns between the pump excitation and probe sensing. The pump beam, modulated by an electro-optical modulator, heats the sample. Upon pump heating, the probe beam, reflected from the metal transducer layer, is collected by a fast-response photodiode for further signal processing with a radio frequency lock-in amplifier. The collected ratio of signals ($-V_{in}/V_{out}$) is compared with a thermal model to extract the thermal properties of samples. Before measurements, thin transducer films of aluminum (Al) were deposited on the surface of end-polished samples by magnetron sputtering. For thermal modeling, the thickness ($h_{Al}$), volumetric heat capacity ($C_{Al}$), thermal conductivity ($\kappa_{Al}$) of the Al transducer, and volumetric heat capacity of samples ($C_{sample}$) are input parameters. Al transducer thickness $h_{Al}$ was measured by picosecond acoustics; the $C_{Al}$ was taken from literature[62]; the $\kappa_{Al}$ was calibrated by the four-point probe method and the Wiedemann-Franz law; the volumetric heat capacity of $sp^3$ amorphous carbon samples were estimated using the specific heat of graphite (0.71 J g$^{-1}$ K$^{-1}$) and diamond (0.50 J g$^{-1}$ K$^{-1}$), mass density and the $sp^3/sp^2$ ratio of samples[63]. Before the TDTR measurement of samples, standard materials (Si, $Al_2O_3$ and $SiO_2$) were also used to calibrate the system using the same measurement parameters (Supplementary Fig. 3).

## Data availability

All data supporting the findings of this study are available within this article and from the corresponding authors upon request. Source data are provided with this paper.

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

## Acknowledgements

This work was supported financially by the National Key R&D Program of China (2018YFA0305900 (B.L.) and 2018YFA0703400 (M.Y.)), the National Natural Science Foundation of China (51822204 (M.Y.), 51320105007 (B.L.), U23A20561 (B.L.), 11634004 (Z.L.), 41902034 (Z.L.), 12104175 (J.D.), 12304015 (Y.S.)), and the China Postdoctoral Science Foundation (2022M720054 (Y.S.), 2023T160257 (Y.S.)).

## Author contributions

B.L. and M.Y. conceived and designed the study. Y.S., Z.L. M.Y., and X.H. synthesized the materials. Y.S. and C.Z. performed the XRD, Raman, UV-vis absorption measurements. Y.S. and M.Y. performed the TEM characterization and analysis. Y.S. performed Vickers hardness measurements. Y.S., J.Z and Z.Z. performed the TDTR measurements and data analysis. Y.S., H.L., L.Yan. R.F., L.Yang. M.Y., J.D. L.F., G.Z., and J.J. performed the synchrotron XRD measurements and data analysis. Y.S., M.Y., B.L., Z.L., and B.S. analyzed the results of data. M.Y., Y.S., B.S., and B.L. wrote the manuscript. All authors discussed the results and contributed to the final manuscript.

## Competing interests

The authors declare no competing interests.
