## [Peer Review File · Nature Communications]

REVIEWER COMMENTS

Reviewer #1 (Remarks to the Author):

Tuning the atomic structure and properties of amorphous material is an interesting but challenging research topic. In this manuscript by Shang et al, the authors report a strategy to tune the local structural ordering of sp³ amorphous carbon by changing the concentration of carbon pentagons and hexagons in the precursor, e.g., by introducing C70. In this way they created a new transparent, nearly pure sp³ amorphous carbon. The obtained amorphous carbon material inherits more hexagonal-diamond structural feature and shows the stronger short/medium-range structural order compared to that synthesized from C60. It also exhibits significantly enhanced thermal conductivity, making a new record for amorphous materials, and higher hardness. This work is seriously performed and well organized. The data is convincing, including many different experimental methods and comprehensive structure and physical properties characterizations.

I believe that the exciting finding of this paper is an advance in the field of amorphous materials and carbon materials, and could attract great interest for the readers of Nature Communications. Therefore, I'd like to recommend it for publication in Nature Communications after the following minor comments/suggestions are addressed.

1. The authors should provide more details on the structure characterizations of initial C70 precursors. In the paper, the author only provided the XRD patterns of C70 precursor (Fig. 1). Raman spectra, for example, could give more helpful atomic and molecule structure information of fullerene, which should be added for comparison.

2. EELS analysis is an effective tool to determine the sp³ content of amorphous carbon materials. However, In Fig. 2, the authors provided all the Raman spectra data of samples AC70-1 to AC70-7, but for the EELS data, the spectra of sample AC70-5 (27GPa 900°C) were not provided. In addition, for the sample AC70-2 synthesized at high temperature (1200°C), XRD and TEM results showed that a small amount of crystalline diamond was formed in this sample (Fig. 1, Fig. 3), but the EELS spectra of this sample still had a small sp² shoulder at 285 eV. Could the author explain what the sp² signal comes from?

3. The high thermal conductivity of sp³ amorphous carbon samples is very interesting, as it not only represents the advanced properties of this new sp³ amorphous carbon, but also reflects their typical structural feature (stronger local structural order). The authors should provide more details about the TDTR measurements. For example, did the authors measure a standard material to calibrate the system using similar parameters? And how reproducible are the test results?

4. The authors claimed that the obtained sp³ amorphous carbon sample is millimeter-sized, but the scale bar was not provided in Fig. 1.

5. All the error bars, such as the hardness, thermal conductivity and sp³ content, should be defined in the figure legend. The abbreviations in Figures, such as a-Ge, DLC etc., should also be defined in the figure legend.

6. Some figures need to be modified. For example, the black font (represent sp³ content) in Figure 1a and the white font in the inserts of Fig. 3a, b, c are too small.

7. A recent paper (Advanced Functional Materials 32, 2203894, 2022) on the structural searching may be cited to improve the argument of formation mechanisms of the amorphous crystal in the current work.

Reviewer #2 (Remarks to the Author):

In this manuscript, the authors report the synthesis, at a pressure of 30 GPa and high temperature, of a millimeter size highly transparent and nearly pure sp³-hybridized bulk amorphous carbon using C70 as a precursor. The authors studied the hybridization and atomic arrangements of the carbon atoms in the resulting materials using a variety of techniques, including XRD, HRTEM, Raman spectroscopy, EELS, PDF etc. as well as characterizing various properties such as thermal conductivity, hardness...

Most recently, research groups have shown that it is possible to produce millimetre-scale forms of disordered diamond at high pressure and high temperature in a large press using fullerenes as the starting material. Further research in this direction is very important from a fundamental and application point of view. The paper is well written, the analyses are serious and accurate. I am glad recommending this paper for publication in Nature Communications journal.

Just a few minor remarks:

1) Abstract : Sentence from row 30 to 33. This amorphous carbon shows more...., stronger ...enhanced ...and highercompared to what ? Please indicate what you are comparing with.

2) Row 125-126: can be the method to calculate sp³ percentage briefly described?

3) In figure 2a please indicate the peak at 285 eV in the EELS spectra , I found difficult to follow the discussion, the various peaks observed in the EELS spectrum of sample AC70-2 (27 GPa, 1200 °C) should also be described. Also it is preferable to follow the same sample order of Figures 1 or to indicate in figure 2 the samples with acronym + P, T synthesis conditions.

4) Row 235: please remember to the reader what is a paracrystalline diamond

5) Row 283: 9.8 N is not the Vickers hardness but the load applied, please correct.

Reviewer #3 (Remarks to the Author):

This work reports on a nearly pure sp^3 -hybridized bulk amorphous carbon by treating C70 at high pressure and high temperature. Compared with the authors' previous work of synthesizing sp^3 amorphous carbon from C60, this work shows that the properties of amorphous carbon can be considerably tuned by using a different precursor, C70. It is interesting that the two amorphous carbon samples (from C70 vs. C60), with highly similar atomic structures and sp^3 fraction, present quite different thermal conductivity. These results suggest that the properties of sp^3 -hybridized amorphous carbon could be precursor sensitive which is rarely observed in other glass/amorphous materials. Hence, these results would be intriguing to the amorphous materials community. There are still some questions and concerns that need to be addressed before this manuscript can be accepted.

1. For the thermal conductivity measurement, what are the sample size and the laser beam size? The samples seem to contain some cracks according to their images shown in Fig.1. Would these cracks affect the measurement results?

2. For AC70-1 and AC3, their difference in $S(q)$ or $G(r)$ is subtle. If the difference in peak intensity is reliable, the peak position should shift accordingly. Subtle changes in $S(q)$ or $G(r)$ sometimes are risky to be attributed to structural differences if the measurements are done on different samples (with different sizes, thicknesses, densities, etc.). Consistent results from various aspects would help to reduce the risk. In Fig. 3, the authors compare the MRO clusters in two samples. How is the "cluster percentage" derived and justified? It would be more convincing if the authors could estimate the errors

in Fig.3f. To support the conclusion of the manuscript, it is also essential to evaluate the uniformity of the MRO cluster distribution by comparing HRTEM images from different sample locations.

3. The reduced PDF $G(r)$ has to equal 0 at $r=0$. It doesn't seem to be the case in Fig. 4b. The PDF data needs to be carefully checked. I suggest the authors show $G(r)$ from $r=0$.

4. In Fig. 4c, have the curves been shifted vertically? If yes, it should be noted in the text. In addition, some relevant references are missing, e.g., DIOPTAS, PDFGETX2, etc.

Reviewer #4 (Remarks to the Author):

This paper reports on the synthesis of amorphous carbon with high concentration of sp^3 bonded carbon (diamond-like) starting from C-70 fullerene precursor. A high concentration of sp^3 bonded carbon and optical transparency is obtained at 30 GPa and 1100 C in sample designated AC70-1.

This paper runs very similar to results reported earlier starting with C-60 precursors in Nature 599, 599–604 (2021) as summarized below. Sample designated as AC-3 in earlier publication using C60 as a precursor.

(1) AC70-1 synthesized at 30 GPa 1100 C similar to AC-3 synthesized at 27 GPa 1000 C.

(2) The EELS spectrum and UV Raman spectrum are basically identical for AC70-1 and AC-3 in Figure 2. The Raman bands designated as "T" and "R" are identical for AC70-1 and AC-3.

(3) The sp^3 diamond content for AC70-1 is 96.2%(error of $\pm 0.9\%$) and sp^3 diamond content for AC-3 is 95.1% (error of $\pm 1.7\%$). This is same diamond content given the experimental uncertainty.

(4) The only difference between the two studies is the thermal conductivity measurements in Figure 4. Here the thermal conductivity data should be compared to measurements on various types of diamond single crystals with nitrogen impurity (Type I- slight yellow) and relatively pure diamond (Type IIa). The optical transparency in their sample AC70-1 and AC-3 tend to show yellow coloration and nitrogen impurity level in these samples should be quantified before comparison.

Overall, there are not enough new results to justify publication in Nature.

Response to the referees:

Reviewer #1 (Remarks to the Author):

Tuning the atomic structure and properties of amorphous material is an interesting but challenging research topic. In this manuscript by Shang *et al.*, the authors report a strategy to tune the local structural ordering of sp^3 amorphous carbon by changing the concentration of carbon pentagons and hexagons in the precursor, e.g., by introducing C_{70} . In this way they created a new transparent, nearly pure sp^3 amorphous carbon. The obtained amorphous carbon material inherits more hexagonal-diamond structural feature and shows the stronger short/medium-range structural order compared to that synthesized from C_{60} . It also exhibits significantly enhanced thermal conductivity, making a new record for amorphous materials, and higher hardness. This work is seriously performed and well organized. The data is convincing, including many different experimental methods and comprehensive structure and physical properties characterizations.

I believe that the exciting finding of this paper is an advance in the field of amorphous materials and carbon materials, and could attract great interest for the readers of Nature Communications. Therefore, I'd like to recommend it for publication in Nature Communications after the following minor comments/suggestions are addressed.

Response: We thank the reviewer for the very positive comments on our manuscript.

Comment 1 - The authors should provide more details on the structure characterizations of initial C_{70} precursors. In the paper, the author only provided the XRD patterns of C_{70} precursor (Fig. 1). Raman spectra, for example, could give more helpful atomic and molecule structure information of fullerene, which should be added for comparison.

Response: According to the reviewer's suggestion, we carried out Raman measurements on the starting C_{70} precursor by using the same laser wavelengths (514 nm) for our amorphous carbon characterization. The recorded Raman spectra are

shown in Figure R1 and Figure 2b, which shows the typical spectroscopic features of C_{70} and is consistent with previous studies [W. Cui *et al.*, *Advanced Materials* 2014, 26, 7257-7263].

Figure R1: Visible (514 nm) Raman spectra of samples recovered from different HPHT conditions and starting C_{70} precursor. AC-3 is the nearly pure sp^3 amorphous carbon synthesized from C_{60} .

Comment 2 - EELS analysis is an effective tool to determine the sp^3 content of amorphous carbon materials. However, In Fig. 2, the authors provided all the Raman spectra data of samples AC70-1 to AC70-7, but for the EELS data, the spectra of sample AC70-5 (27 GPa 900 °C) were not provided. In addition, for the sample AC70-2 synthesized at high temperature (1200 °C), XRD and TEM results showed that a small amount of crystalline diamond was formed in this sample (Fig. 1, Fig. 3), but the EELS spectra of this sample still had a small sp^2 shoulder at 285 eV. Could the author explain what the sp^2 signal comes from?

Response: According to the reviewer's suggestion, we added the EELS data of sample AC70-5 in Figure 2a for comparison. We also calculated the percentage of sp^3 content for AC70-5, which is $87.4 \pm 5.5\%$ (Figure R2, Figure 2a), similar with the AC70-6 sample, which is consistent with our Raman spectroscopic characterization (Figure R1). For the AC70-2 sample, the visible small ramp at the region of π^* in the

EELS spectrum should be originated from the dangling sp^2 bonds and amorphous defects covering on the diamond nanocrystals surface, as evidenced by our HRTEM observation (see Figure R3).

Figure R2: EELS spectra of samples recovered from different HPHT conditions. AC-3 is the nearly pure sp^3 amorphous carbon synthesized from C_{60} .

Figure R3: HRTEM images of AC70-2 sample. It can be seen that the sp^2 amorphous defects covering on the surface of diamond nanocrystals.

Comment 3 - The high thermal conductivity of sp^3 amorphous carbon samples is very interesting, as it not only represents the advanced properties of this new sp^3

amorphous carbon, but also reflects their typical structural feature (stronger local structural order). The authors should provide more details about the TDTR measurements. For example, did the authors measure a standard material to calibrate the system using similar parameters? And how reproducible are the test results?

Response: We thank the reviewer for the suggestions about the TDTR measurements. Before the TDTR measurement, we calibrate the system by using standard material (Si, Al₂O₃ and SiO₂) (Figure R4). We can see the measured thermal conductivity values of standard material are well consistent with those reported in literatures (Figure R4b). This ensured the reliability and accuracy of our thermal conductivities measured from amorphous samples when similar measurement parameters are used.

Figure R4: **a** The measured TDTR data for the standard Si, Al₂O₃ and SiO₂ samples. **b** The measured thermal conductivities of standard materials compared with literature data.

About the reproducibility of the thermal conductivity test, for each sample, we selected several (>3) smooth and crack-free regions randomly for TDTR measurement. As shown in Figure R5, we measured four points randomly on the sample AC70-1, and the measured TDTR signals are almost the same. The results from the four measurements gave an average thermal conductivity of $36.3 \text{ W m}^{-1} \text{ K}^{-1}$ for AC70-1 and a standard deviation of $2.2 \text{ W m}^{-1} \text{ K}^{-1}$.

We have added these data into our Supplementary Information. Please see Supplementary Figures 3, 4 and 5.

Figure R5: The measured TDTR signals (**a-d**) and the corresponding four test locations for AC70-1 sample (**e-h**). The bright spot is the laser focus position.

Comment 4 - The authors claimed that the obtained sp^3 amorphous carbon sample is millimeter-sized, but the scale bar was not provided in Fig. 1.

Response: According to the reviewer's suggestion, we added the scale bar of samples in Figure 1.

Comment 5 - All the error bars, such as the hardness, thermal conductivity and sp^3 content, should be defined in the figure legend. The abbreviations in figures, such as a-Ge, DLC etc., should also be defined in the figure legend.

Response: According to the reviewer's suggestion, we have defined all the error bars and abbreviations in the figure legend.

Comment 6 - Some figures need to be modified. For example, the black font (represent sp^3 content) in Figure 2a and the white font in the inserts of Fig. 3a, b, c are too small.

Response: We thank the reviewer for the suggestion. The mentioned figures have been modified according to the suggestion.

Comment 7 - A recent paper (Advanced Functional Materials 32, 2203894, 2022) on the structural searching may be cited to improve the argument of formation mechanisms of the amorphous crystal in the current work.

Response: We have cited the paper in the discussion part of our revised manuscript, please see page 14 and line 323.

Reviewer #2 (Remarks to the Author):

In this manuscript, the authors report the synthesis, at a pressure of 30 GPa and high temperature, of a millimeter size highly transparent and nearly pure sp^3 -hybridized bulk amorphous carbon using C_{70} as a precursor. The authors studied the hybridization and atomic arrangements of the carbon atoms in the resulting materials using a variety of techniques, including XRD, HRTEM, Raman spectroscopy, EELS, PDF etc. as well as characterizing various properties such as thermal conductivity, hardness...

Most recently, research groups have shown that it is possible to produce millimetre-scale forms of disordered diamond at high pressure and high temperature in a large press using fullerenes as the starting material. Further research in this direction is very important from a fundamental and application point of view. The paper is well written, the analyses are serious and accurate. I am glad recommending this paper for publication in Nature Communications journal.

Response: We thank the reviewer for the very positive comments on our manuscript.

Just a few minor remarks:

Minor comment 1 - Abstract: Sentence from row 30 to 33. This amorphous carbon shows more..., stronger ...enhanced ...and highercompared to what? Please indicate what you are comparing with.

Response: We are sorry for this unclear description. The structure and properties of the current amorphous carbon are compared with the sp^3 amorphous carbon synthesized from C_{60} . We thus revised the statement to “This amorphous carbon shows more hexagonal-diamond-like clusters, stronger short/medium-range structural order, and significantly enhanced thermal conductivity ($36.3 \pm 2.2 \text{ W m}^{-1} \text{ K}^{-1}$) and higher hardness ($109.8 \pm 5.6 \text{ GPa}$) compared to that synthesized from C_{60} .”

Minor comment 2 - Row 125-126: can be the method to calculate sp^3 percentage briefly described?

Response: We used the peak-ratio method developed by Berger *et al.* to calculate the sp^3 percentage of amorphous carbon [S. D. Berger *et al.*, *Philosophical Magazine Letters* 1988, 57, 285-290]. A standard EELS spectrum of fully sp^2 glassy carbon is shown in Figure R6, which shows two characteristic features: the $1s-\pi^*$ transition peak at around 285 eV and the $1s-\sigma^*$ transition peak at around 291 eV. According to the method described by Berger *et al.*, it is assumed the density of π and σ states is proportional to the ratio of integrated areas under π^* and σ^* peaks. So when glassy carbon (GC) is used as standard sp^2 carbon materials, the sp^3 percentage of our synthesized amorphous carbon (AC) can be calculated according to the formula:

$$sp^3\% = 1 - \frac{\left[\frac{area(\pi^*)}{area(\pi^* + \sigma^*)} \right]_{AC}}{\left[\frac{area(\pi^*)}{area(\pi^* + \sigma^*)} \right]_{GC}}$$

Note that before we calculated the sp^3 percentage, the background of recorded spectrum was removed by using the Digital Micrograph software and the π^* and σ^* peaks were fitted by using a Gaussian function [S. D. Berger *et al.*, *Philosophical Magazine Letters* 1988, 57, 285-290; Y. F. Su *et al.*, *Microscopy and Microanalysis* 2016, 22, 666-672.].

Figure R6: EELS spectra of standard sp^2 glassy carbon.

We have added a brief description about the method in the revised version, please see page 5, lines 128-130.

Minor comment 3 - In Figure 2a please indicate the peak at 285 eV in the EELS spectra, I found difficult to follow the discussion, the various peaks observed in the EELS spectrum of sample AC70-2 (27 GPa, 1200 °C) should also be described. Also it is preferable to follow the same sample order of Figures 1 or to indicate in Figure 2 the samples with acronym + P, T synthesis conditions.

Response: Thanks for the suggestions. The sp^3 concentration in our amorphous carbon samples are high (>75%), and thus the $1s-\pi^*$ peak is very weak in the EELS spectra, which all appear as a small shoulder peak in the energy range of ~282-286 eV, as shown in Figure R7. We have added the indication of the energy locations of $1s$ to π^* transition (labeled $1s-\pi^*$) and the $1s$ to σ^* transition (labeled $1s-\sigma^*$) in the revised Figure (Figure 2a, Figure R7).

For the sample AC70-2 synthesized at 27 GPa, 1200 °C, both XRD measurement and HRTEM observations show that it contains a certain amount of nanocrystalline diamond (Figure 1 and Figure 3b). The EELS spectrum of AC70-2 thus exhibits spectroscopic features of nanocrystalline diamond, in which the visible small ramp at the region of π^* (282-286 eV) was originated from the sp^2 dangling bonds and defects on nanodiamond surface, the peaks in the energy range of 290-310 eV were originated from $1s$ to σ^* transition, a dip at ~302 eV is attributed to the second absolute band gap of diamond [P. J. Pauzauskie *et al.*, *Proceedings of the National Academy of Sciences* 2011, 108, 8550-8553; M. C. Guenette *et al.*, *Diamond and Related Materials*, 2013, 34, 45-49]. The weak broad peak at ~320 eV is originated from the multiple-scattering involving a $1s$ to σ^* transition and an $\sigma-\sigma^*$ interband transition (plasmon peak) [P. E. Batson *et al.*, *Physical Review Letters*, 1979, 42, 893]. The revised Figure 2 also presents results in the same sample order of Figures 1.

Figure R7: EELS spectra of samples recovered from different HPHT conditions. AC-3 is the nearly pure sp^3 amorphous carbon synthesized from C_{60}

Minor comment 4 - Row 235: please remember to the reader what is a paracrystalline diamond

Response: We have added a brief description of paracrystalline diamond in our revised manuscript. Please see page 11, lines 246-249.

Minor comment 5 - Row 283: 9.8 N is not the Vickers hardness but the load applied, please correct.

Response: Thanks for the reminder. We have revised the statement to “The nearly pure sp^3 amorphous carbon inherits more hexagonal-diamond structural feature and shows higher local structural order than that synthesized from C_{60} , which exhibits a much higher thermal conductivity and a higher Vickers hardness at a load of 9.8 N (Fig. 6c) than any other amorphous solid known up to now.”

Reviewer #3 (Remarks to the Author):

This work reports on a nearly pure sp^3 -hybridized bulk amorphous carbon by treating

C₇₀ at high pressure and high temperature. Compared with the authors' previous work of synthesizing *sp*³ amorphous carbon from C₆₀, this work shows that the properties of amorphous carbon can be considerably tuned by using a different precursor, C₇₀. It is interesting that the two amorphous carbon samples (from C₇₀ vs. C₆₀), with highly similar atomic structures and *sp*³ fraction, present quite different thermal conductivity. These results suggest that the properties of *sp*³-hybridized amorphous carbon could be precursor sensitive which is rarely observed in other glass/amorphous materials. Hence, these results would be intriguing to the amorphous materials community. There are still some questions and concerns that need to be addressed before this manuscript can be accepted.

Response: We thank the reviewer for the very positive comments on our manuscript.

Comment 1 - For the thermal conductivity measurement, what are the sample size and the laser beam size? The samples seem to contain some cracks according to their images shown in Fig.1. Would these cracks affect the measurement results?

Response: Our thermal conductivity measurement was using a non-contact time-domain thermoreflectance (TDTR) technique, which is a pump-probe optical method to characterize thermal properties, especially for the samples in micrometer size [J. Zhu *et al.*, *Nanoscale and Microscale Thermophysical Engineering* 2017, 21, 177–198; C. Yuan *et al.*, *Applied Physics Letters*, 2021, 119, 133902]. The sample for TDTR measurement is 1 mm in diameter. Before TDTR measurement, the sample was polished and the polished smooth area is about 500 μm in diameter. To avoid the effects of cracks and sample boundary, the laser with a small spot size (spot radius, $r=11.6$ μm) (Figure R8b) was used in our thermal conductivity measurement, and TDTR signals only reflect the thermal properties of the laser focus areas. Thus the thermal conductivity properties of amorphous carbon samples can be obtained by measuring the local areas where the sample is dense and crack-free (Figure R8a). Furthermore, in order to avoid measurement errors, we selected several (at least 3) smooth and crack-free regions randomly for TDTR measurement, and the thermal conductivity value is given by averaging the results of several measurements (Figure

R5, Supplementary Figure 4, 5).

We have added these details of TDTR test to the Supplementary Information, please see Supplementary Note 1 and Supplementary Figures 4 and 5.

Figure R8: **a** The optical image of the smooth area of sample for TDTR measurements, the bright spot is the laser focus position. **b** The beam offset signals used for measuring the beam spot size.

Comment 2 - For AC70-1 and AC-3, their difference in $S(Q)$ or $G(r)$ is subtle. If the difference in peak intensity is reliable, the peak position should shift accordingly. Subtle changes in $S(Q)$ or $G(r)$ sometimes are risky to be attributed to structural differences if the measurements are done on different samples (with different sizes, thicknesses, densities, etc.). Consistent results from various aspects would help to reduce the risk.

Response: We thank the referee for this interesting comment.

For AC70-1 and AC-3, although their difference in $S(Q)$ or $G(r)$ is subtle, the peak intensity of FSDP in $S(Q)$ and first shell r_1 and second shell r_2 in $G(r)$ are all higher for AC70-1 (Figure R9a, c). This indicates that the local structural order of AC70-1 is higher than that of AC-3. Moreover, according to the reviewer's suggestions, we further analyzed the change of the peak position in $S(Q)$ or $G(r)$ for the two samples. Figure R9b shows the magnified FSDP in $S(Q)$, although the peak position difference is very small, we can see the peak position of FSDP for AC70-1 is clearly higher than that of AC-3. This indicates a slightly increased density for

AC70-1 sample. For $G(r)$, we also found that the first r_1 and second r_2 peak for AC70-1 both shift slightly to the left and are closer to the standard tetrahedral atomic configuration of crystalline diamond (r_1 , 1.54 Å, r_2 , 2.515 Å) (Figure R9d). These results further prove the higher density and local structural order for AC70-1, which is consistent with the results of $S(Q)$. Furthermore, the stronger local structural order also can be evidenced by our HRTEM analysis. HRTEM observations indicate that the fraction of local ordered regions for AC70-1 is higher than that of AC-3 (Figure 3f, Figure R10c). And the fraction of HD-like clusters in AC70-1 is about twice that in AC-3 (Figure 3f, Figure R10c). These results further confirm that the local structural order of AC70-1 is stronger than that of AC-3.

To give further support, we use a high photon flux beamline (BL13HB) in Shanghai Synchrotron Radiation Facility (SSRF) for PDF measurements. Note that high-energy and high-flux synchrotron X-rays (X-Ray cross section, ~1mm) can completely cover and penetrate our carbon samples, so the subtle differences in the size and thickness of the samples should not affect our PDF characterizations. As described above, the peak position of $S(Q)$, which represents the d -spacing of lattice-like fringes and is related to the density of sample, slightly shifts to the right for AC70-1, further supporting a slightly increased density for AC70-1 sample compared to that of AC-3.

We have added the discussion of the difference of peak position in $S(Q)$ and $G(r)$ for AC70-1 and AC-3 in our revised manuscript, please see page 9, lines 213-227.

Figure R9: **a** Structure factor $S(Q)$ of AC70-1 and AC-3 samples. **b** The magnified FSDP in $S(Q)$. **c** The reduced PDF profiles, $G(r)$, of AC70-1 and AC-3 samples. **d** The magnified first r_1 and second r_2 peak in $G(r)$. The red arrows in **a**, **b**, **c** and **d** indicate the peak intensity increase for AC70-1. The black arrow in **b** indicates the peak position shift to the right for AC70-1. The black arrows in **d** indicate the peak position shift to the left for AC70-1.

In Fig. 3, the authors compare the MRO clusters in two samples. How is the “cluster percentage” derived and justified? It would be more convincing if the authors could estimate the errors in Fig. 3f. To support the conclusion of the manuscript, it is also essential to evaluate the uniformity of the MRO cluster distribution by comparing HRTEM images from different sample locations.

Response: We thank the referee for this serious comment. The percentage of MRO clusters was quantitatively estimated through a simple statistical analysis method. As

shown in Figure R10a, b, the diamond-like MRO clusters can be identified and distinguished because of their lattice-like fringes with two different intersecting angles, corresponding to the (111) plane of cubic diamond (CD) and the (100) plane of hexagonal diamond (HD). The HD-like MRO clusters was marked by yellow square and the CD-like was marked by yellow circle. To describe the structural order in atomic scale, we first randomly selected a large area and then carefully counted the area percentage of HD-like and CD-like MRO clusters in this area (Figure R10d-i).

According to the reviewer's suggestion, statistical analysis of cluster percentage was performed in more different areas for the two samples (AC70-1 and AC-3) (Figure R10d-i). The results show that the total areal fraction of the local ordered regions is $38.1 \pm 3.8\%$ for AC70-1, which is higher than that of AC-3 ($31.6 \pm 2.3\%$) (Figure R10c). It is also found that the fraction of HD-like clusters in AC70-1 ($10.1 \pm 1.0\%$) is about twice that in AC-3 ($4.3 \pm 1.0\%$). These results confirm that the local structural order of AC70-1 is stronger than that of AC-3, agreeing well with our PDF results.

The new data was added in the Supplementary information (Supplementary Figure 1) and the related description and discussion have been added in our revised version, please see page 8, lines 190-196.

Figure R10: **a** The typical inverse FFT image of AC70-1, the regions marked with yellow square represent the HD-like MRO cluster. **b** The typical inverse FFT image of AC-3, the regions marked with yellow circle represents the CD-like MRO cluster. **c** Content statistics of diamond-like MRO clusters in AC70-1 and AC-3. Error bars indicate statistical analysis for three different areas, standard deviations. **d-f** Content statistics of diamond-like MRO clusters in AC70-1. **g-i** Content statistics of diamond-like MRO clusters in AC-3.

Comment 3 - The reduced PDF $G(r)$ has to equal 0 at $r=0$. It doesn't seem to be the case in Fig. 4b. The PDF data needs to be carefully checked. I suggest the authors show $G(r)$ from $r=0$.

Response: We thank the referee for this comment.

Yes, according to the physical definition of PDF, the reduced PDF $G(r)$ has to equal 0 at $r=0$. And in our $G(r)$ data, $G(r)$ indeed equal 0 at $r=0$ (Figure R9c). However, it should be kept in mind that the experimental $G(r)$ is generated from reciprocal space scattering information which is limited by an inevitable momentum cutoff Q and the data quality of high- Q region. The attainable Q_{\max} particularly affects the shape of low- r region in $G(r)$ according to the Fourier transform formula:

$$G(r) = \frac{2}{\pi} \int_{Q_{\min}}^{Q_{\max}} Q[S(Q) - 1] \sin(Qr) dQ$$

So, it is difficult to achieve a perfect baseline in the realistic experimental measurement [G. S. E. Antipas *et al.*, *Methods X* 2019, 6, 601-605; M. Moseler *et al.*, *Physical Review letters*, 2005, 94, 165503]. This also happens to our $G(r)$ data in the low- r region (0-1Å) (Figure R9c), because of the limited Q_{\max} (14 Å⁻¹) and data quality of $S(Q)$ in the high- Q region. In order to obtain higher-quality PDF data, we performed PDF measurement again at the Shanghai Synchrotron Radiation Facility (SSRF) (see our response to comment 2 above). In this measurement, we collected high-quality $S(Q)$ data with higher Q_{\max} up to 16 Å⁻¹ (Figure R11a) by reducing the distance between the sample and detector. The new PDF results are shown in Figure R11c. We can see that the low- r region (0-1Å) in $G(r)$ is reasonable, achieving a relatively good baseline. Furthermore, our new $S(Q)$ data again demonstrates that the peak intensity of FSDP for AC70-1 is obviously higher than that of AC-3 (Figure R11a, b). And the peak intensity of first shell r_1 and second shell r_2 in $G(r)$ for AC70-1 is also higher than those of AC-3 (Figure R11c, d). These results further confirmed the local structural order of AC70-1 is higher than that of AC-3.

The new data and the related description and discussion have been added in our revised version, please see pages 9-11, lines 213-244. And the old PDF data was moved to the Supplementary Information (Supplementary Figure 2).

Figure R11: **a** The new structure factor $S(Q)$ data with a Q_{\max} of 16 \AA^{-1} for AC70-1 and AC-3 samples. **b** The magnified FSDP in $S(Q)$. **c** The new reduced PDF profiles, $G(r)$, of AC70-1 and AC-3 samples. **d** The magnified first r_1 and second r_2 peak in $G(r)$. The red arrows in **a**, **b**, **c** and **d** indicate the peak intensity increase for AC70-1. The black arrow in **b** indicates the peak position shift to the right for AC70-1. The black arrows in **d** indicate the peak position shift to the left for AC70-1.

Comment 4 - In Fig. 4c, have the curves been shifted vertically? If yes, it should be noted in the text. In addition, some relevant references are missing, e.g., DIOPTAS, PDFGETX2, etc.

Response: According to the reviewer's suggestion, the vertical shift of curves was noted in the figure legend. And the relevant references were also added. Please see page 16, lines 399-401.

Reviewer #4 (Remarks to the Author):

This paper reports on the synthesis of amorphous carbon with high concentration of sp^3 bonded carbon (diamond-like) starting from C_{70} fullerene precursor. A high concentration of sp^3 bonded carbon and optical transparency is obtained at 30 GPa and 1100 °C in sample designated AC70-1.

This paper runs very similar to results reported earlier starting with C_{60} precursors in Nature 599, 599–604 (2021) as summarized below. Sample designated as AC-3 in earlier publication using C_{60} as a precursor.

Response: We thank the reviewer for the comments on our manuscript.

Comment - 1, 2 and 3 -AC70-1 synthesized at 30 GPa 1100 °C similar to AC-3 synthesized at 27 GPa 1000 °C. The EELS spectrum and UV Raman spectrum are basically identical for AC70-1 and AC-3 in Figure 2. The Raman bands designated as "T" and "R" are identical for AC70-1 and AC-3. The sp^3 diamond content for AC70-1 is 96.2% (error of $\pm 0.9\%$) and sp^3 diamond content for AC-3 is 95.1% (error of $\pm 1.7\%$). This is same diamond content given the experimental uncertainty.

Response: In the current paper we focused on the amorphous carbon synthesized from C_{70} , while the AC-3 amorphous carbon from C_{60} is added for comparison. The motivation of this work is to investigate the effect of 5- and 6- carbon rings in carbon precursors on the microstructures of the synthesized amorphous carbon under high pressure and high temperature. We discovered that, despite their close synthesis conditions, AC70-1 from C_{70} (30 GPa 1100 °C) and AC-3 from C_{60} (27 GPa 1000 °C), samples exhibit obvious/significant differences in structure and properties. By using advanced characterization techniques such as HRTEM and synchrotron PDF, we carefully investigated the structural difference of the two amorphous carbon samples. The structure factor $S(Q)$ data shows that AC70-1 exhibits a more intense FSDP compared to that of AC-3 (Figure R11a, b, Figure 4a, b) and the PDF data also indicates that the peak intensity of the first shell (r_1) and second shell (r_2) are both stronger for AC70-1 (Figure R11c, d, Figure 4c, d). Our HRTEM analysis also indicates that the areal percentage of local ordered regions of AC70-1 is higher than

that of AC-3 (Figure R10c, Figure 3f, Supplementary Fig. 1c), and the fraction of HD-like clusters in AC70-1 is about twice that in AC-3. These results strongly suggest that the two samples exhibit obvious structural difference, compared to AC-3, AC70-1 inherits more hexagonal-diamond structural feature, showing a higher fraction of hexagonal-diamond-like clusters and stronger short/medium-range structural order.

Such obvious structural difference for the two samples also leads to a significant properties difference. Compared to AC-3 sample, the stronger local structural order for AC70-1 caused significantly enhanced thermal conductivity ($36.3 \pm 2.2 \text{ W m}^{-1} \text{ K}^{-1}$) (Figure 4f, Figure 6b) and higher hardness ($109.8 \pm 5.6 \text{ GPa}$) (Figure 5a, Figure 6c), setting a new record in thermal conductivity of amorphous materials.

Comment 4 - The only difference between the two studies is the thermal conductivity measurements in Figure 4. Here the thermal conductivity data should be compared to measurements on various types of diamond single crystals with nitrogen impurity (Type I- slight yellow) and relatively pure diamond (Type IIa). The optical transparency in their sample AC70-1 and AC-3 tend to show yellow coloration and nitrogen impurity level in these samples should be quantified before comparison.

Response: We thank the reviewer for the suggestions. Indeed, our current amorphous carbon from C_{70} has different thermal conductivities from that by C_{60} . According to the reviewer's suggestion, we first investigated the thermal conductivities of diamond single crystals with different concentrations of nitrogen impurity. Previous literatures show that the thermal conductivity of relatively pure Type IIa diamond single crystals ($\sim 2500 \text{ W m}^{-1} \text{ K}^{-1}$ at 300 K) is much higher than that of Type I diamond with higher N concentration ($\sim 700\text{-}1500 \text{ W m}^{-1} \text{ K}^{-1}$ at 300 K) (Figure R12) [J. R. Olson *et al.*, *Physical Review B*, 1993, 47, 14850]. This can be well explained by the scattering of phonons at point imperfections [J. R. Olson *et al.*, *Physical Review B*, 1993, 47, 14850; W. Kaiser *et al.*, *Physical Review*, 1959, 115, 857]. However, for our sp^3 amorphous carbon, AC70-1 exhibits a thermal conductivity of $36.3 \pm 2.2 \text{ W m}^{-1} \text{ K}^{-1}$, higher than that of AC-3 ($26.0 \pm 1.3 \text{ W m}^{-1} \text{ K}^{-1}$). This is due to the stronger local

structural order for AC70-1, as described in our manuscript.

Figure R12: Thermal conductivity of natural type-IIa diamond and type-Ia diamond with different N concentration [J. R. Olson *et al.*, *Physical Review B*, 1993, 47, 14850].

To explore the possible effects of nitrogen impurity on thermal conductivity of sp^3 amorphous carbon, as suggested by the referee, we carefully examined the EELS spectra, EDS spectra and PL spectra of AC70-1 and AC-3 samples (Figure R13). We can see that all those characterizations do not show any evidence for or trace of the presence of N-related signals in the two samples. Our EDS spectra also show that the two samples contain 100% carbon, with no other impurity element detectable (see Figure R13c, d); our PL spectra of the samples also do not show any N-related emission (Figure R13a) [O. R. Rubinas *et al.*, *Results Phys.* 2021, 21, 103845]. Thus, the higher thermal conductivity for AC70-1 is contributed from its stronger structural order, as described in our manuscript.

Figure R13: PL, EELS and EDS spectra of AC70-1 and AC-3. **a** PL spectra excited by 514 nm laser. The green and yellow solid lines indicate possible positions for N-center-related emission. **b** EELS spectra for AC-3 sample, no signal at 401 eV from nitrogen was observed. **c and d** Typical EDS spectra of AC-3 and AC70-1 sample. No impurity can be detected.

REVIEWERS' COMMENTS

Reviewer #1 (Remarks to the Author):

The revised version can be accepted for publication in the current version.

Reviewer #2 (Remarks to the Author):

The authors carefully responded to all the reviewers' comments and criticisms and modified figures and texts accordingly. They also performed new analyses at the request of some reviewers. I am very pleased to recommend the article for publication in Nature Communications . No further reviews are demanded from my side

Reviewer #3 (Remarks to the Author):

The supplementary experimental data and the response have addressed my comments and concerns. I am glad to recommend this paper for potential publication in Nature Communications. However, I'd like to add one more last comment on the PDF data interpretation. For $S(q)$ and $G(r)$, the peak intensity is sensitive to the experimental conditions and data analysis, such as the experimental coverage of q or q cut-off and the background subtraction. In comparison, the peak position is usually more reliable. Therefore, I suggest the authors clarify and emphasize this point to avoid any misunderstanding of readers.

Reviewer #4 (Remarks to the Author):

The authors have clearly demonstrated in revised manuscript and their response that there is no nitrogen contamination in the synthesized samples of sp^3 amorphous carbon from C70 precursors. This strengthened their arguments about improved thermal conductivity. I recommend publication.

Response to the referees:

Reviewer #1 (Remarks to the Author):

The revised version can be accepted for publication in the current version.

Response: We appreciate the reviewer for his/her recommendation of publication of our manuscript.

Reviewer #2 (Remarks to the Author):

The authors carefully responded to all the reviewers' comments and criticisms and modified figures and texts accordingly. They also performed new analyses at the request of some reviewers. I am very pleased to recommend the article for publication in Nature Communications. No further reviews are demanded from my side.

Response: We appreciate the reviewer for his/her very positive comments and recommendation of publication of our manuscript.

Reviewer #3 (Remarks to the Author):

The supplementary experimental data and the response have addressed my comments and concerns. I am glad to recommend this paper for potential publication in Nature Communications. However, I'd like to add one more last comment on the PDF data interpretation. For $S(Q)$ and $G(r)$, the peak intensity is sensitive to the experimental conditions and data analysis, such as the experimental coverage of Q or Q cut-off and the background subtraction. In comparison, the peak position is usually more reliable. Therefore, I suggest the authors clarify and emphasize this point to avoid any misunderstanding of readers.

Response: We appreciate the reviewer for his/her recommendation of publication of our manuscript. According to the kind suggestion of the reviewer, we added the description about the PDF data interpretation in **Methods**, and clarified the effects of the experimental conditions and data analysis, such as the experimental coverage of Q or Q cut-off and the background subtraction, on the peak intensity of $S(Q)$ and $G(r)$. Please see page 17, lines 419-427.

Reviewer #4 (Remarks to the Author):

The authors have clearly demonstrated in revised manuscript and their response that there is no nitrogen contamination in the synthesized samples of sp^3 amorphous carbon from C_{70} precursors. This strengthened their arguments about improved thermal conductivity. I recommend publication.

Response: We appreciate the reviewer for his/her very positive comments and recommendation of publication of our manuscript.